# Function-oriented design of robust metal cocatalyst for photocatalytic hydrogen evolution on metal/titania composites

Dong Wang[1 ✉] & Xue-Qing Gong [1]

While the precise design of catalysts is one of ultimate goals in catalysis, practical strategies often fall short, especially for complicated photocatalytic processes. Here, taking the hydrogen evolution reaction (HER) as an example, we introduce a theoretical approach for designing robust metal cocatalysts supported on $TiO_2$ using density functional theory calculations adopting on-site Coulomb correction and/or hybrid functionals. The approach starts with clarifying the individual function of each metal layer of metal/$TiO_2$ composites in photocatalytic HER, covering both the electron transfer and surface catalysis aspects, followed by conducting a function-oriented optimization via exploring competent candidates. With this approach, we successfully determine and verify bimetallic Pt/Rh/$TiO_2$ and Pt/Cu/$TiO_2$ catalysts to be robust substitutes for conventional Pt/$TiO_2$. The right metal type as well as the proper stacking sequence are demonstrated to be key to boosting performance. Moreover, we tentatively identify the tunneling barrier height as an effective descriptor for the important electron transfer process in photocatalysis on metal/oxide catalysts. We believe that this study pushes forward the frontier of photocatalyst design towards higher water splitting efficiency.

[1] Key Laboratory for Advanced Materials and Joint International Research Laboratory for Precision Chemistry and Molecular Engineering, Feringa Nobel Prize Scientist Joint Research Center, Centre for Computational Chemistry and Research Institute of Industrial Catalysis, School of Chemistry and Molecular Engineering, East China University of Science and Technology, 130 Meilong Road, Shanghai 200237, China. ✉email: wangd@ecust.edu.cn

Titanium dioxide ($TiO_2$) is an important material for photocatalytic water splitting[1–3], on which loading of proper cocatalysts (usually metals) is routinely used to enhance the hydrogen evolution reaction (HER) performance[4,5]. As photoelectrons are first generated in the bulk region of $TiO_2$ after photon excitation[6–8], good metal/titania catalysts should qualify at least two aspects of (i) efficient electron transfer across the interface (from $TiO_2$ to the metal) and (ii) rapid $H_2$ production on metal surface[9,10]. Noble metals such as Pt, Pd, and Rh are able to meet both requirements, as benefited from their large work functions (or low Fermi levels)[4,11] and suitable Gibbs adsorption energies of H atom ($\Delta G_H$)[12,13], and thus they were frequently used in photocatalytic HER[5,11]. In particular, Pt is known as the optimal material for catalyzing HER[4,5,11], but how to reduce the catalyst cost and/or increase the activity has aroused significant interests in both academia and industry.

One potential solution is to properly regulate the interface structure. Since the interface bridges the oxide and metals, its structure as well as the metal-support interaction (MSI)[14], on one hand controls the adhesive contact strength and the electron transfer process, and on the other hand affects the surface reaction activity owing to the induced charge redistributions[15–18]. Many modification techniques, such as morphology engineering[19,20], particle size control[21,22], alloying[23,24] etc., are all able to alter the interface properties and furthermore the photocatalytic performance (albeit uncertainty in promotion or not). Nevertheless, in addition to trial-and-error experiments, strategic knowledges about how to conduct rational design of metal/oxide catalysts and where to start are still open questions. Besides, given that the $\Delta G_H$ is generally regarded as an effective descriptor signifying surface HER activity[12,13], whether there also exists some physicochemical properties that could directly scale the electron transfer efficiency remains elusive yet.

Intriguingly, Umezawa and colleagues[25] investigated electronic structures of metal/$TiO_2$ composites (metal = Pt, Pd, Au) and found that the MSI-induced charge redistribution is largely confined within the first contact metal layer and drops quickly to the second and the third layer. Similar phenomena have also been observed at other metal/oxide interfaces[26,27]. These findings imply that each metal layer at the interface is unique and may exhibit distinct catalytic behaviors compared with other layers or bulk metals (i.e., layer-dependent catalytic function). Moreover, our previous study on the size effect of supported Pt cocatalysts demonstrated that smaller one-Pt-layer clusters have good electron transfer character but low surface H–H coupling activity, whereas larger multilayer particles show an opposite trend with limited electron transfer efficiency[28]. It was thus anticipated that (i) the interface first metal layer might be related to the electron transfer process, while the exterior layer to surface catalytic reactions; (ii) further, by optimizing the individual function of each layer, the overall performance could be improved.

Herein, we report a function-oriented design of efficient metal/$TiO_2$ catalysts for photocatalytic HER. Differing from the precise size control of Pt nanoparticles to reach a balance between the electron transfer and surface catalysis[28], we achieved in theory excellent performance in both aspects (rather than making compromises), by means of alloying robust electron transfer and surface reaction components in a right combination sequence. Furthermore, once the optimal alloying patterns were determined, it is technically feasible to fabricate them in experiments perhaps via the atomic layer deposition (ALD)[29,30] or underpotential deposition[31,32] techniques. For example, Moffat and colleagues[23] reported the precise control of the structure and quantity of deposited metals even in the range of monolayer level, whereas Chen and Bent[33] realized the area-selective ALD by selectively pre-adsorbing a proper molecule (e.g., 1-octadecene, octadecyltrichlorosilane), which resists the attachment of deposition precursors on the substrate.

## Results

**Exploring robust electron transfer materials**. As Pt is highly effective for surface HER and thus the best candidate for constructing the exterior metal layer[12,13], the first priority is to find excellent materials for the interface electron transfer layer. Herein, we adopted our previous approach of calculating the intrinsic electron transfer (IET) energy[28], defined by the energy change of moving an excess electron from $TiO_2$ bulk to supported metal particles in the absence of surface adsorbates, to compare the electron transferring ability among candidate metals. The initial state of a localized electron at a Ti site ($Ti^{3+}$) in the bulk region of $TiO_2$ and the final state of delocalized electron densities on supported metal particles were confirmed by calculating the site-projected magnetic moments and Bader charges (see details in "IET energy calculation" in Methods)[15,28]. It should be mentioned that, since there are many possible electron trapping sites in bulk $TiO_2$, we have calculated at least three different Ti sites for each metal/$TiO_2$ composite and the reported energies were the averaged value. Eventually, the IET results can be obtained by comparing the total energy along the electron transfer pathway, as summarized in Supplementary Table 1.

In Fig. 1, we assessed the electron transferring ability of 10 candidate metals (Supplementary Fig. 1) by calculating the IET energies. It is noteworthy that other transition metals (first ~ third periods) are generally too active with electron affinity < 1 eV[34] and can readily cause interface distortion (e.g., Hf and Ta) and/or oxide reduction (e.g., Ti, Mn, Co, Nb, Mo, and W) upon deposition (Supplementary Fig. 2)[35], and thus were not considered in the screening scope. It was found that noble metals in the VIII group (except Ru) are able to promote the directional electron transfer (namely, from bulk $TiO_2$ to the metal) with exothermic IET energies, but Ag and Au turn to be bad choices, consistent with experimental observations[5,11]. In particular, the screening results (using the $M_8/TiO_2$ model) suggest that Rh, with strong IET energy of −0.27 eV, might be more efficient than Pt (−0.17 eV) in facilitating the directional electron transfer, but the main drawback is again its high price. Encouragingly, cheap metal Cu also shows an impressive IET energy of −0.15 eV close to Pt, serving as a promising electron transfer material. Moreover, the excellent electron transferring ability of Rh and Cu was further confirmed by using relatively larger $M_{19}/TiO_2$ models, obtained via ab initio molecular dynamics simulations (AIMD; Supplementary Fig. 3). Therefore, we theoretically proposed two bimetallic models, constructed by one layer of noble Rh or cheap Cu directly adhering to $TiO_2$ and a second layer of Pt capping on the top, to be competitive substitutes (either in benefits or costs) of the conventional Pt for photocatalytic $H_2$ production.

**Verifying the effectiveness of proposed catalysts**. By utilizing geometry features of the stable $Pt_{13}/TiO_2$ composite (obtained via

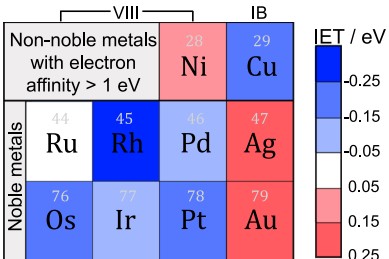

**Fig. 1 Schematic of calculated IET energies in the presence of different metals on $TiO_2$.** Colors from red to blue indicate gradually enhancing IET energies (using $M_8/TiO_2$ or $M_7/TiO_2$ models) and the details are shown in Supplementary Table 1 and Supplementary Fig. 1.

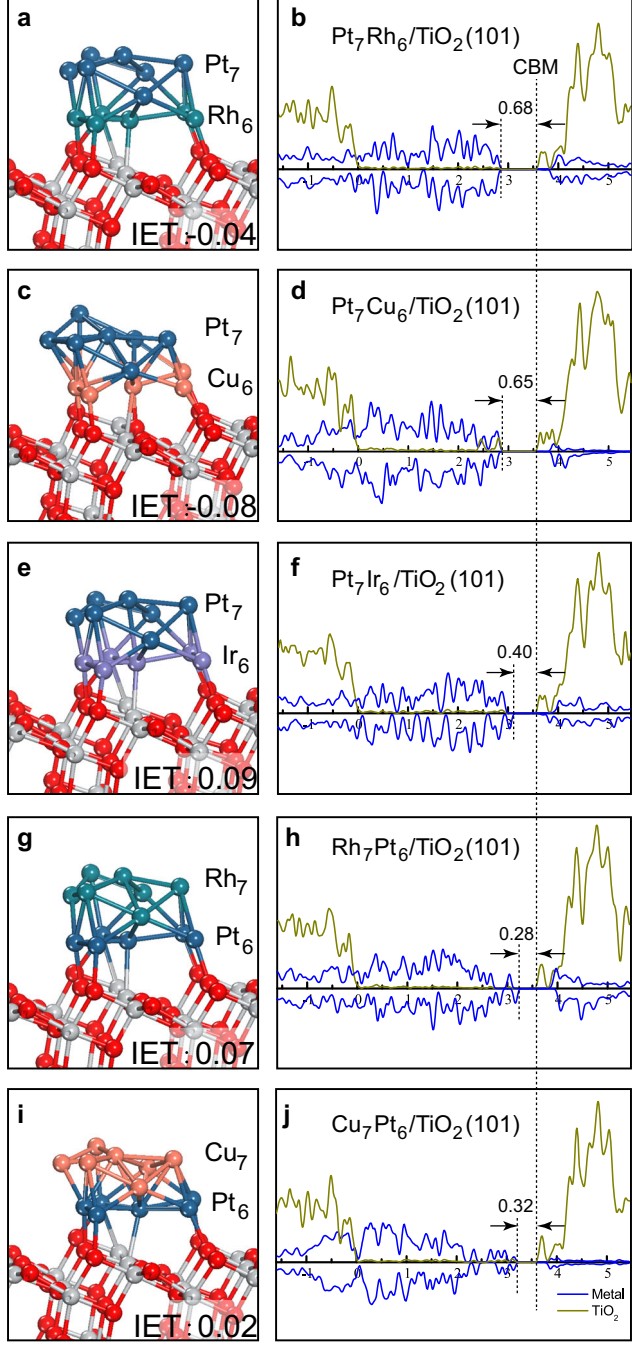

**Fig. 2 Calculated structures, IET energies, and density of states for bimetallic metal/TiO₂ composites.** a, b for $Pt_7Rh_6/TiO_2(101)$, c, d for $Pt_7Cu_6/TiO_2(101)$, e, f for $Pt_7Ir_6/TiO_2(101)$, g, h for $Rh_7Pt_6/TiO_2(101)$, and i, j for $Cu_7Pt_6/TiO_2(101)$, respectively. The DOS (density of states; using the hybrid HSE06 functional) for $M_{13}$ cluster and for $TiO_2$ surface are represented by blue and yellow-green curves, respectively. The valance band edge of $TiO_2$ is uniformly aligned to zero in DOS, and the vertical dot lines indicate the $TiO_2$ CBM and the $E_f$ position, respectively. Calculated IET energies are also given in eV.

AIMD simulations; Supplementary Fig. 4) and replacing the first interface layer of six Pt atoms with Rh or Cu, we determined the bimetallic structure of $Pt_7Rh_6/TiO_2(101)$ and $Pt_7Cu_6/TiO_2(101)$ via unconstrained relaxations. No obvious structure changes were observed for the $Pt_7Rh_6/TiO_2$ (Fig. 2a) relative to that of $Pt_{13}/TiO_2$, indicating a lattice-matched alloying between Pt and Rh, whereas

noticeable distortions for the $Pt_7Cu_6/TiO_2$ (Fig. 2c) can be attributed to the distinct atomic radius difference (~7%)[36]. Further, IET energy calculations showed that the directional electron transfer (confirmed by charge density plots in Supplementary Fig. 5) indeed gets promoted for both the $Pt_7Rh_6/TiO_2$ and $Pt_7Cu_6/TiO_2$ composites, being −0.04 and −0.08 eV, respectively, as compared with the endothermic 0.05 eV for $Pt_{13}/TiO_2$[15,28]. Such promotion is consistent with the downward shift of the Fermi level ($E_f$), from 0.45 eV below $TiO_2$ conduction band minimum (CBM) for $Pt_{13}/TiO_2$[28] to ~0.65 eV (see Fig. 2b, d), considering that a relatively lower energy level is beneficial to accept electrons. It is also necessary to address that the geometry of deposited metals has significant impacts on their physicochemical properties (Supplementary Fig. 6), signifying the importance of determining stable metal/oxide interfaces on reliable calculation results.

In addition, we also picked a less effective electron transfer metal of Ir for fair comparison, using similar approaches above. Consistently, the $Pt_7Ir_6/TiO_2$ composite (Fig. 2e) was found to be no better than $Pt_{13}/TiO_2$ with an endothermic IET energy of 0.09 eV, verifying the effectiveness of our screening approach. Furthermore, we investigated the electron transfer for models with inverted metal stacking sequence, namely the $Rh_7Pt_6/TiO_2$ and $Cu_7Pt_6/TiO_2$ composites (Fig. 2g, i). As expected, the electron transfer process turns to be retarded with endothermic IET energies and up-shifted $E_f$ (Fig. 2g–j). Despite minor oscillations induced by the upper Rh or Cu layers, the electron transfer ability of the two inverted models generally resembles that of $Pt_{13}/TiO_2$. These results revealed evidently the electron transfer function of the first interface metal layer, and demonstrated that not only the right metal type (Rh or Cu) but also proper stacking sequence (Pt/Rh/TiO₂, Pt/Cu/TiO₂) are essential to achieve improved electron transfer efficiency.

Next, it is necessary to verify whether the catalytically efficient HER continues to proceed on the exterior Pt layer of the bimetallic composites. Considering the diverse $\Delta G_H$ results on different surface sites (Supplementary Table 2) and also the volcano-shaped $\Delta G_H$ ~ activity relationship[9,10], we uniformly selected two of the most active sites with $\Delta G_H$ approaches zero for computing the H–H coupling barrier ($E_a^{coup}$) and determining the HER activity of the alloy models. In Fig. 3a, b, one can see that both $Pt_7Rh_6/TiO_2$ and $Pt_7Cu_6/TiO_2$ give moderate hydrogen adsorption strength close to the volcano peak of $\Delta G_H = 0$ eV[9,10]. Moreover, we determined the coupling barrier $E_a^{coup}$ to be 0.71 and 0.69 eV for $Pt_7Rh_6/TiO_2$ and $Pt_7Cu_6/TiO_2$ (Fig. 3d, e), respectively, being even slightly lower than the value of 0.75 eV on $Pt_{13}/TiO_2$[28]. It is noteworthy that the coupling state on $Pt_7Rh_6/TiO_2$ shows quite late transition structure resembling the $H_2$ adsorption configuration, which was further confirmed by using the climbing image nudged elastic band method (CI-NEB; Supplementary Fig. 7)[37]. Apparently, the HER activity on these alloy cocatalysts could be higher than extended Pt surfaces, owing to large $E_a^{coup}$ of 0.88 and 1.07 eV on Pt(111) and Pt(100), respectively[38].

Hence, by integrating good surface catalytic activity of Pt and robust electron transfer ability of Rh or Cu, the alloyed Pt/Rh or Pt/Cu cocatalyst breaks the intrinsic constraints between the two key photocatalytic factors for mono-type materials (e.g., Pt), and could in principle achieve superior performance than conventional $Pt/TiO_2$. It is worth noting that this strategy resembles the concept of using multiphase materials to break the scaling constraint between chemisorption energies of key intermediates (e.g., C and O in CO oxidation) in heterogenous catalysis[39] and also offers probabilities to approach the global activity maximum in photocatalysis. In particular, we underline the striking results of $Pt/Cu/TiO_2$ with enhanced photocatalytic performance but reduced catalyst cost as compared to pure Pt cocatalysts. This is further validated by using a trilayer $Pt_6Cu_{13}/TiO_2$ model with much lower Pt content, where

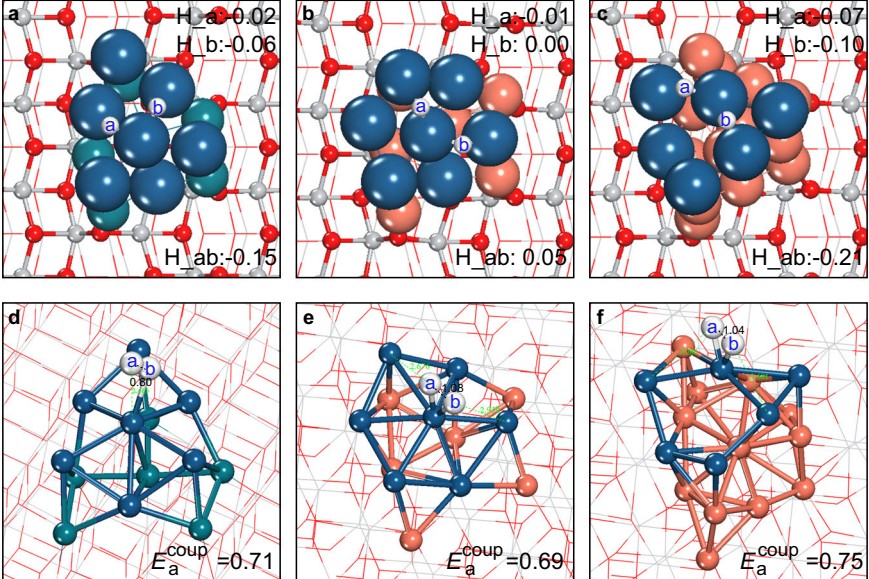

**Fig. 3 Calculated structures and energies for H adsorption and H–H coupling on proposed metal/TiO₂ composites. a–c** H adsorption at two reactive sites (labeled as a and b) and **d–f** transition states of H–H coupling on the Pt₇Rh₆/TiO₂, Pt₇Cu₆/TiO₂, and Pt₆Cu₁₃/TiO₂ composites, respectively. Gibbs adsorption energies for the independent adsorption of H atom on each of the two reactive sites (H_a and H_b), the co-adsorption of two H atoms (H_ab), as well as the H–H coupling barrier $E_a^{coup}$, are given in eV.

suitable $\Delta G_H$ and $E_a^{coup}$ are still well conserved (Fig. 3c, f). Furthermore, the Pt/Cu/TiO₂ composite was validated to be fairly stable via long-time AIMD simulations (>20 ps; Supplementary Fig. 8). Overall, TiO₂ supported Cu nanoparticles coated with a thin film of Pt skin are theoretically predicted to be efficient yet inexpensive catalysts for photocatalytic HER.

**Generalized correlations**. Finally, given the crucial role of electron transfer in photocatalysis, one may ask what physical properties might be related with the electron transfer efficiency across the interface. Here we calculated the effective potential, which represents the carrier interaction with other electrons in the system and the external electrostatic field[40,41], of relevant metal/TiO₂ composites using the HSE06 functional (Supplementary Table 1). In particular, the tunneling barrier height ($\Phi_{TB}$), defined as the potential difference between the peak at the interface and TiO₂ slab (Fig. 4a), indicates the minimum energy cost of directional electronic injection via quantum tunneling[40,41]. Intriguingly, the IET energies was found to correlate well with the $\Phi_{TB}$, which although is typically mentioned in van der Waals junctions (e.g., metal/MoS₂ interfaces)[42], among a wide range of metal/TiO₂ composites (Fig. 4b) holding short interface distances (Supplementary Table 1). Furthermore, despite the significant impact of particle size on IET energies as previously discovered[28], the linear relationship between the IET energy and the $\Phi_{TB}$ was found irrelevant to either the metal type (indicated by different colors) or particle size (different shapes in Fig. 4b). We notice that the number of sampling points might still be limited, but the good correlation coefficient in a wide sampling range encourages to regard such correlation as a general trend (at least for supported metals on TiO₂), which is beyond the considered metal/TiO₂ models in this work. As far as we know, this is the first piece of report on the electron transfer descriptor for photocatalytic reactions on metal/oxide catalysts. Considering the significance of electron transfer efficiency in photocatalysis, this finding could possibly promote future photocatalysts design via coordinating conventional catalytic descriptors (e.g., $\Delta G_H$ for HER) and the electron transfer properties $\Phi_{TB}$.

## Discussion

In summary, we decomposed the complicated photocatalytic HER (after photoexcitation) into elementary processes of interface electron transfer and surface catalytic reactions in this work. For metal/TiO₂ composites, the interfacial first metal layer was evidenced to be responsible for collecting photoelectrons from TiO₂, while the exterior layer is largely related to catalysing surface HER. Regarding particularly the former, we theoretically identified Rh and Cu to be robust electron transfer materials, which, after alloying with the optimal HER catalyst of Pt, give rise to bimetallic Pt/Rh/TiO₂ and Pt/Cu/TiO₂ as excellent substitutes for the conventional Pt/TiO₂. Our results demonstrated clearly the importance of both the right metal type and the proper stacking sequence in improving the performance of alloyed materials. We furthermore generalized a linear correlation between IET energies and the tunneling barrier height $\Phi_{TB}$ of metal/TiO₂ composites, paving the way to the rational design of highly-efficient and cost-effective photocatalysts, for instance, the proposed Pt/Cu/TiO₂ for photocatalytic hydrogen production.

## Methods

**DFT calculations**. All density functional theory (DFT) calculations were performed using the Vienna Ab-initio Simulation Package (VASP) with the spin-polarization being considered. The DFT functional was utilized at the Perdew–Burke–Ernzerhof level. The project-augmented wave method was used to represent the core-valence electron interaction. The valence electronic states were expanded in plane wave basis sets with energy cutoff at 450 eV. The transition states were searched using a constrained optimization scheme and were verified when (i) all forces on atoms vanish and (ii) the total energy is a maximum along the reaction coordination but a minimum with respect to the rest of the degrees of freedom. The force threshold for the transition state search and structural optimization was 0.05 eV/Å. The dipole correction was applied throughout the calculations to take the polarization effect into account.

For the TiO₂ system, we have demonstrated previously[43,44] that the DFT + U method can yield similar structures and energies as those from the hybrid HSE06 functional. Here we mainly applied the DFT + U method in computing the thermodynamic properties (e.g., adsorption energy and energy barriers), where the on-site Coulomb correction was set on Ti 3d orbitals with an effective U value of 4.2 eV as suggested in literature works[45,46]. To produce electronic structure properties more accurately (e.g., band gap, band edge position, and effective potential), we further performed hybrid HSE06 calculations utilizing the DFT + U geometry, in which the electronic minimization algorithm was set as ALGO = All with a very soft augmentation charge (PRECFOCK = Fast). The lattice parameters and band gap of bulk anatase TiO₂ determined via the HSE06 functional ($a = b = 3.766$, $c = 9.448$ Å;

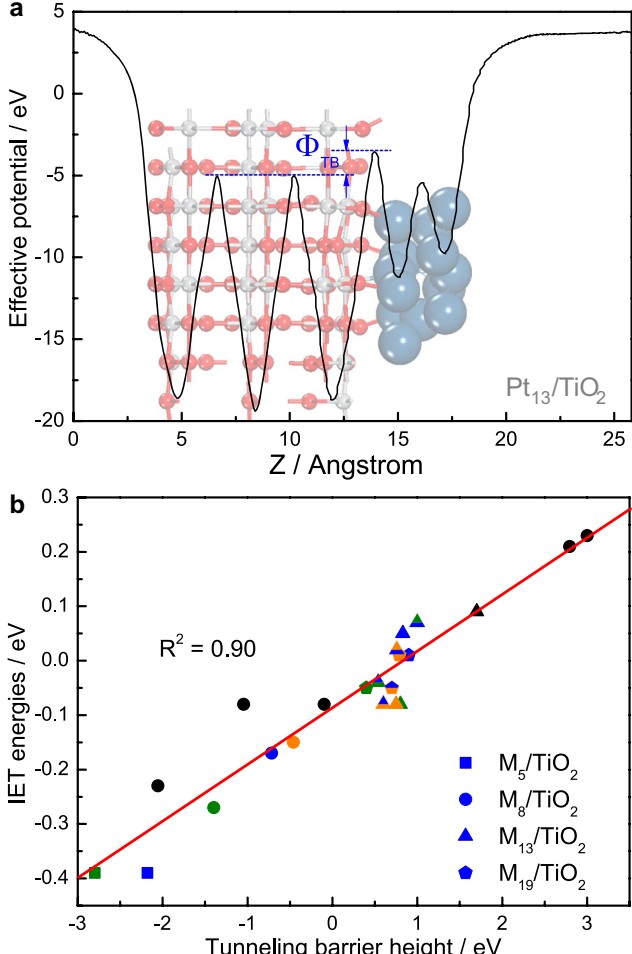

**Fig. 4 Calculated effective potential curve and the generalized linear correlation. a** Schematic of the effective potential (black curve) and $\Phi_{TB}$ (indicated by blue arrows) of $Pt_{13}/TiO_2$ using the hybrid HSE06 functional, as well as **b** the linear correlation between IET energies and $\Phi_{TB}$ on a wide range of metal/TiO$_2$ composites. While Pt, Rh, and Cu are highlighted in blue, olive, and orange color, the $M_5/TiO_2$, $M_8/TiO_2$, $M_{13}/TiO_2$, and $M_{19}/TiO_2$ are illustrated by solid square, dot, triangle, and pentagon in plot **b**, respectively. Symbols filled with mixed colors represent alloyed metal particles, where the stacking pattern from surface to the interface is consistent with the top-down coloring sequence.

$E_g = 3.31$ eV) agree well with the experimental values ($a = b = 3.776$, $c = 9.486$ Å; $E_g = \sim3.2$ eV).

The effective potential was calculated by including (i) the Hartree potential caused by the mean-field electrostatic interaction, (ii) the exchange-correlation potential due to the quantum mechanical nature of the electrons, and (iii) other external electrostatic interactions in the system. Technically, this can be done by invoking the LVTOT parameter in the input file using the VASP program (version 5.4 or later).

The adsorption energy of hydrogen atom ($E_{ad}^H$) is defined as the energy difference after and before the adsorption with respect to the gas phase H$_2$ molecule according to $E_{ad}^H = E(total) - E(surface) - 1/2E(H_2)$, where $E(surface)$, $E(H_2)$, and $E(total)$ are the energies for the clean surface, H$_2$ molecule in the gas phase, and hydrogen atom adsorbed on the surface, respectively. It is noteworthy that $\Delta G_H$ is obtained by adopting the entropy correction proposed by Norskov et al.[12,13] according to $\Delta G_H = E_{ad}^H + 0.24$. The more negative the $E_{ad}^H$ (or $\Delta G_H$) is, the more strongly the H atom binds on surface.

**Model construction for metal/TiO$_2$ composites.** To properly accommodate the metal cluster, we utilized a relative large supercell of anatase (101) surface, that is a $p$ (2 × 3) three-Ti-layer slab (72 Ti and 36 O atoms) with a vacuum thickness of 16 Å. We have checked the electron transfer energetics by enlarging the slab thickness to four-Ti-layer models, which produce similar results as that from three-layer calculations

(Tables S1 and S2 in ref. [15]). It is noteworthy that all the metal/TiO$_2$ composites containing non-Pt metals were constructed by utilizing geometry features of the optimal $Pt_{13}/TiO_2$ or $Pt_8/TiO_2$, or $Pt_7/TiO_2$ composites, determined via extensive AIMD simulations as shown in Supplementary Fig. 4, and replacing Pt with other metals by needs. Both the conjugate gradient and Quasi-Newton optimization methods were used to determine the stable structures.

Regarding the optimal structure of $Rh_{19}/TiO_2$, $Cu_{19}/TiO_2$ and bimetallic $Pt_7Cu_6/TiO_2$ composites, AIMD simulation was performed to investigate their structural evolutions. The simulation was carried out in the canonical NVT ensemble employing Nosé–Hoover thermostats. The temperature was set at 450 K that is taken from the temperature of hydrothermal treatment commonly used in experiments[21,47] and the time step was 1 fs. More than 20 ps AIMD simulations were performed (Supplementary Figs. 3 and 8) and from the equilibrated trajectory we selected structural snapshots in every 2 ps interval and fully optimized them until all forces diminish. The optimal composite structure can be determined as the lowest-energy configuration.

**IET energy calculation.** The extra photoelectron in the system was simulated by adding an excess electron into the supercell as common practice[8,46,48]. We have also previously validated the approach by comparing the electron transfer results with those obtained by introducing an additional H atom on the opposite layer of TiO$_2$ slab in charge-neutral systems (Table S1 in ref. [28].). The localization of the electron on a particular Ti site of TiO$_2$ can be initially configured and followed by DFT + U electronic structure optimizations. Initial magnetic moments on each atom are usually necessary in the input settings, although they will be optimized during the calculation. None of any constraints on the distribution/population of spin-polarized charges were imposed during the optimization.

To calculate the IET energy, the initial state is set as a localized electron at a Ti site in the bulk region of TiO$_2$. The as-formed Ti$^{3+}$ cation shows distinctive features relative to the regular lattice Ti$^{4+}$ and one can distinguish them easily as follows. (i) Geometry structures: the electron localization (generation of Ti$^{3+}$ cation) is accompanied with the distinct elongations of Ti-O bonds by the outward movement of surrounding lattice O, consistent with literation reports[8,48]. (ii) Electronic structure analysis: the position of electron localization can be further confirmed by calculating the site-projected magnetic moments (~0.8 $\mu_e$) and Bader charges (negatively charged with ~0.4 |e| relative to lattice Ti$^{4+}$ in bulk TiO$_2$), as visualized by the spin density plots in Supplementary Fig. 5. Whereas for the final state, the electron delocalization on metal particles can also be confirmed by (i) the absence of Ti$^{3+}$ cation in TiO$_2$ but (ii) the increased electron quantities of ~0.4 |e| on metals (relative to the charge-neutral metal/TiO$_2$ model) via Bader charge analysis. It should also be noted that, our research focused on the post-photoexcitation electron migration process, which can be approximately regarded as a ground-state event, and therefore no excited states were simulated in calculating the IET energies.

## Data availability
The data that support the results of this study are available from the corresponding author upon reasonable request.

## Code availability
The commercialized VASP code was used only and no custom codes were involved in this work.

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

## Acknowledgements

We acknowledge the financial support from the National Key Research and Development Program of China (2018YFA0208602) and National Natural Science Foundation of China (21903025 and 21825301).

## Author contributions

D.W. conceived and conducted the research and wrote the paper. X.Q.G. participated in results discussion and manuscript modification. All authors commented on the manuscript.

## Competing interests

The authors declare no competing interests.
