## [Peer Review File · Nature Communications]

REVIEWER COMMENTS

Reviewer #1 (Remarks to the Author):

Using density functional theory computations, the authors focused on the hydrogen evolution reaction and introduce a theoretical approach for designing robust metal cocatalysts supported on TiO₂. Based on the approach, they further verified bimetallic catalysts to be robust substitutes for conventional Pt/TiO₂. These results are timely, intriguing and may help to develop novel photocatalysts with high efficiency and low cost. Overall, I would recommend the publication of the manuscript after the following minor comments are properly addressed.

1. The authors calculated the intrinsic electron transfer (IET) energies from bulk TiO₂ to the metal, with the aim to estimate the electron transferring ability of the ten candidate metals. It would be more readable if the authors could define the concept/meaning of IET energies in the main text. It is not clear how did they determine the values (and may be compare with the results from other literature) in Figure 1.
2. In Figure 4, the tunneling barrier height was denoted to indicate the minimum energy cost of directional electronic injection via quantum tunneling. However, according to the recent literatures, the tunneling barrier is often characterized by two parameters, i.e. the tunneling barrier height and width [see, for example, JACS 2019, 141, 3110; 2D Mater. 2017, 4, 025116]. The authors should comment on this point.
3. It would be good if the authors could address how to fabricate these bimetallic structures in experiments.

Reviewer #2 (Remarks to the Author):

This theoretical paper addresses the important problem of identifying new materials to improve the efficiency of the photocatalytic process of hydrogen production in HER reactions. Here the authors have performed an extended series of first-principles calculations on mixed and pure M13 clusters supported on the surface of anatase TiO₂ to correlate some fundamental properties of the catalysts with their catalytic properties. The main concept is that the reaction depends critically on what is called "intrinsic electron transfer" energy. This is defined as the energy required to move one (photo-excited) electron from the bulk of TiO₂ to the supported cluster; it can be negative or positive, depending on the composition of the supported cluster.

While the idea is nice, the way the results are presented is unsatisfactory. There are two main problems with the way the paper is written: lack of details and clarity about the way many properties are calculated; abuse of strong unsupported statements.

About the first point, the way IET energy is computed needs to be better specified. In the supplementary material this is defined as the difference between an initial and a final state with the unpaired electron in two different positions, a bulk Ti ion, and the supported cluster. How convergence on the two states is obtained? It is mentioned that an electron has been added to the supercell (which thus became charged, with the problem of using a neutralizing background of charge). The ground state is thus doublet. How the excited state is computed? Furthermore, given the critical importance of the IET in the process, this needs to be well described in the main text, not only in the supplementary. An easy criticism is that for a M13 cluster an extremely large number of isomers exists. To what extent the results depend on the shape (and size!) of the cluster? I do not expect this to be without effect. I know that the answer to this question is almost impossible, but at least the problem needs to be mentioned and addressed in a convincing way.

When H is adsorbed on the supported cluster, again there is a huge number of possible isomers. How is this specific isomer selected? To what extent the results are depending on the arbitrary choice of using these specific sites?

The correlation shown in Figure 4 is very interesting, but the data are vague. I count about 20 points in the plot. Are these the structures reported in Table S1? If yes, it is clear that the tunneling barrier height is strongly dependent on the cluster size: in Pt5/TiO₂ is -2.18 eV, in Pt13/TiO₂ it becomes +0.05 eV. Same metal, completely different tunnel barrier height. Not surprising, as we are in the non-scalable regime of cluster properties. Again, to what extent the results presented depend on the particular choice of the cluster size? AND CLUSTER SHAPE?

About the strong statements:

"we pioneeringly identified the tunneling barrier height as an effective electron transfer descriptor for photocatalytic reactions..". In general should not be the authors that define their work pioneering, and furthermore there is no photocatalytic reaction studied in this paper. The sentence needs to be reformulated.

The claim of different HER performance is based on the calculation of barriers for the H-H coupling that differ by 0.02 eV, see p. 5 (0.69 eV versus 0.71 eV). This is well below the intrinsic limitations of the methods used. Here the impression is that the authors are making too strong claims.

To some extent this is true also for the effect of E_f which goes from 0.45 eV below the CB in Pt13/TiO₂ to 0.65 eV for the mixed metal clusters. It is in the right direction, but not a huge change.

On the technical side, the spin densities in Figure S3 are either blue or yellow. If this indicates up and down spins, it also means that the solutions are highly spin contaminated. Some comment is in order. In summary, a paper based on a nice idea which is not sufficiently supported and validated by the data presented.

Reviewer: 1

Comments:

“Using density functional theory computations, the authors focused on the hydrogen evolution reaction and introduce a theoretical approach for designing robust metal cocatalysts supported on TiO₂. Based on the approach, they further verified bimetallic catalysts to be robust substitutes for conventional Pt/TiO₂. These results are timely, intriguing and may help to develop novel photocatalysts with high efficiency and low cost. Overall, I would recommend the publication of the manuscript after the following minor comments are properly addressed.

1. The authors calculated the intrinsic electron transfer (IET) energies from bulk TiO₂ to the metal, with the aim to estimate the electron transferring ability of the ten candidate metals. It would be more readable if the authors could define the concept/meaning of IET energies in the main text. It is not clear how did they determine the values (and may be compare with the results from other literature) in Figure 1.”

Response: We thank the referee for the positive comments and valuable suggestions. As this question was also commented by another referee, please refer to the response to the comment 2 of referee 2 for details. Accordingly, we have added a paragraph to explain the IET energy calculation in the revised manuscript (page 3 line 13-25 in the text).

“2. In Figure 4, the tunneling barrier height was denoted to indicate the minimum energy cost of directional electronic injection via quantum tunneling. However, according to the recent literatures, the tunneling barrier is often characterized by two parameters, i.e. the tunneling barrier height and width [see, for example, JACS 2019, 141, 3110; 2D Mater. 2017, 4, 025116]. The authors should comment on this point.”

Response: We agree with the referee that the tunneling barrier height and width are two key parameters characterizing the tunneling barrier typically mentioned for vdW junctions.^{1,2} However, regarding the metal/TiO₂ composites, the way of the directional electron transfer across the interface is not limited to the quantum tunneling. Differing from the vdW junction with large interface distance ($> 3.0 \text{ \AA}$),² the metal/TiO₂ composite explicitly forms chemical bonds between metals and TiO₂ with short interface distances (Table R1), giving rise to a more common way of electron transfer via polaron hopping (primarily through thermal motion).³ Accordingly, Figure 4 was intended to show that the IET energetics was found to correlate well with the tunneling barrier height among a wide range of metal/TiO₂ composites, rather than to underline the quantum tunneling way of electron transfer. Thus, it is not routinely required to cover both the tunneling barrier height and width as we were not discussing the quantum tunneling issue. Nevertheless, the decoupling of tunneling barrier width from IET energetics in the metal/TiO₂ system is probably because the metal-TiO₂ interface distance is usually very small (1.1~1.6 \AA), which leads to even smaller width of the square potential barrier that might be comparable among various metal/TiO₂ interfaces. Note that this is also the reason why we made the statement as follows ‘Moreover, we pioneeringly identified the tunneling barrier height as an effective electron transfer

descriptor for photocatalytic reactions on metal/oxide catalysts.’ in the original abstract. We have added a note on this issue and incorporated Table R1 in the revised manuscript (page 7 line 4-7 from the bottom in the text and supplementary Table 1).

Table R1 Computed IET energies, tunneling barrier height Φ_{TB} (via hybrid HSE06 functional), and metal-TiO₂ interface distance (the vertical distance between metals and TiO₂) of considered metal/TiO₂ composites.

Composites	Φ_{TB} / eV	IET energy/eV	Distance / Å	Composites	Φ_{TB} / eV	IET energy/eV	Distance / Å
Pt ₅ /TiO ₂	-2.18	-0.39	1.11	Cu ₇ Pt ₆ /TiO ₂	0.76	0.02	1.56
Pt ₈ /TiO ₂	-0.72	-0.17	1.24	Cu ₁₉ /TiO ₂	0.79	0.01	1.51
Pt ₁₃ /TiO ₂	0.83	0.05	1.56	Pt ₆ Cu ₁₃ /TiO ₂	0.70	-0.05	1.57
Pt ₁₉ /TiO ₂	0.90	0.01	1.62	Ir ₈ /TiO ₂	-1.05	-0.08	1.28
Rh ₅ /TiO ₂	-2.80	-0.39	1.10	Pt ₇ Ir ₆ /TiO ₂	0.01	0.09	1.52
Rh ₈ /TiO ₂	-1.40	-0.27	1.19	Os ₈ /TiO ₂	-2.06	-0.23	1.20
Rh ₁₃ /TiO ₂	0.80	-0.08	1.48	Pd ₈ /TiO ₂	-0.10	-0.08	1.24
Pt ₇ Rh ₆ /TiO ₂	0.54	-0.04	1.54	Pd ₁₃ /TiO ₂	1.70	0.09	1.60
Rh ₇ Pt ₆ /TiO ₂	1.00	0.07	1.56	Ag ₇ /TiO ₂	3.00	0.23	1.96
Rh ₁₉ /TiO ₂	0.40	-0.05	1.62	Au ₇ /TiO ₂	2.79	0.21	1.89
Cu ₈ /TiO ₂	-0.46	-0.15	1.18	Ni ₈ /TiO ₂	/	0.09	1.02
Cu ₁₃ /TiO ₂	0.75	-0.08	1.52	Ru ₈ /TiO ₂	/	0.01	1.10
Pt ₇ Cu ₆ /TiO ₂	0.60	-0.08	1.44				

“3. It would be good if the authors could address how to fabricate these bimetallic structures in experiments.”

Response: We thank the referee for the valuable suggestion. Since we are not experts in experiments, the following discussions are simply rough ideas. As far as we know, there are two common methods to precisely lay a foreign metal monolayer on the substrate, namely, the atomic layer deposition (ALD; in vapor phase)^{4,5} and the underpotential deposition (UPD; in liquid phase)^{6,7} techniques. Particularly regarding Pt deposition, a third method called self-terminating rapid electrodeposition process developed by Moffat et al. also enables sequential deposition of thin Pt films of desired thickness.⁸ Therefore, we suggest to prepare Cu nanoparticles supported on TiO₂ first via conventional hydrothermal synthesis, and then adopt one of the three methods above to deposit a thin Pt layer capping on the top. The key challenge is to realize the area-selective deposition that protects bare TiO₂ surface from being covered by Pt. One possible solution is to find a proper molecule (e.g. octadecyltrichlorosilane) that pre-adsorbs on TiO₂ and resists the attachment of deposition precursors (e.g. PtCl₄²⁻, CH₃C₅H₄Pt(CH₃)₃).⁹ As suggested by the referee, changes are made in the revised manuscript to address this issue (page 3 line 2-9 in the text).

Reviewer: 2

Comments:

"1. This theoretical paper addresses the important problem of identifying new materials to improve the efficiency of the photocatalytic process of hydrogen production in HER reactions. Here the authors have performed an extended series of first-principles calculations on mixed and pure M13 clusters supported on the surface of anatase TiO₂ to correlate some fundamental properties of the catalysts with their catalytic properties. The main concept is that the reaction depends critically on what is called "intrinsic electron transfer" energy. This is defined as the energy required to move one (photo-excited) electron from the bulk of TiO₂ to the supported cluster; it can be negative or positive, depending on the composition of the supported cluster.

While the idea is nice, the way the results are presented is unsatisfactory. There are two main problems with the way the paper is written: lack of details and clarity about the way many properties are calculated; abuse of strong unsupported statements."

Response: We thank the referee for the positive comments and valuable suggestions, which contribute to a stronger version of the revised manuscript. Detailed responses are given in below.

"2. About the first point, the way IET energy is computed needs to be better specified. In the supplementary material this is defined as the difference between an initial and a final state with the unpaired electron in two different positions, a bulk Ti ion, and the supported cluster. How convergence on the two states is obtained? It is mentioned that an electron has been added to the supercell (which thus became charged, with the problem of using a neutralizing background of charge). The ground state is thus doublet. How the excited state is computed? Furthermore, given the critical importance of the IET in the process, this needs to be well described in the main text, not only in the supplementary."

Response: This work was conducted on the basis of two pieces of our previous works, in which we studied the mechanism of electron transfer from TiO₂ to metal cocatalysts¹⁰ and the size effect of Pt nanoparticles¹¹ for photocatalytic HER on metal/TiO₂ composites. Previously, we have developed an effective approach to realize and confirm the localization of an electron on a particular Ti site of TiO₂ or electron delocalization on the metal, as detailed in the revised manuscript (section 3 in Methods.). Regarding the former, the as-formed Ti³⁺ cation shows distinctive features relative to the regular lattice Ti⁴⁺, and we can distinguish them easily as follows. (i) Geometry structures: the electron localization (generation of Ti³⁺ cation) is accompanied with the distinct elongation of Ti-O bonds by the outward movement of surrounding lattice O, consistent with literature reports.^{12,13} (ii) Electronic structure analysis: the position of electron localization can be further confirmed by calculating the site-projected magnetic moments ($\sim 0.8 \mu_B$) and Bader charges (negatively charged with $\sim 0.4 |e|$ relative to lattice Ti⁴⁺ in bulk TiO₂), as visualized by the spin density plots in supplementary Fig. 5. While for the latter, the electron delocalization on metal particles can also be confirmed by i) the absence of Ti³⁺ cation in TiO₂ but ii) the increased electron quantities of $\sim 0.4 |e|$ on metals (relative to the charge-neutral metal/TiO₂ model) via Bader charge analysis.

Regarding the calculation of electron transfer energies, our projects primarily focused on the post-photoexcitation process, which can be rationalized as follows. Photogenerated electrons are first produced in the bulk region of TiO₂ after photon excitation,^{12,14,15} and become self-trapped on Ti cations to yield Ti³⁺ within a few picoseconds.^{16,17} In order to trigger surface reactions, they have to move away from TiO₂ bulk and arrive at the metal cocatalyst on the surface, which thus brings a certain amount of excess electrons in the near-surface region (excluding the electron-hole recombination). Such post-photoexcitation electron migration process can be approximately regarded as a ground-state event, and therefore no excited states were simulated in calculating the IET energies. In addition, we agree with the referee that the introduced excess electron may cause unphysical interaction resulting from the presence of background counter charge. However, we have previously compared the electron transfer results with those obtained by introducing an additional H atom on the opposite layer of TiO₂ slab (a charge-neutral system), which produces a protonated O_{2c} and an additional electron trapped at a specific Ti ion.^{10,11} The minor energy difference of ≤ 0.04 eV between two procedures of introducing $1e^-$ or 1H (Table S1 in ref. 11) thus credits the effectiveness of our approach (at least from the energetics aspect), as also widely used in literature reports.^{12,13,18}

We thank the referee for the valuable suggestion. Accordingly, we have added a paragraph to explain the IET energy calculation (page 3 line 13-25 in the text) and incorporated relevant discussions in the revised manuscript (page 9 line 1-5 from the bottom and page 10 line 1-6 in the text).

“3. An easy criticism is that for a M₁₃ cluster an extremely large number of isomers exists. To what extent the results depend on the shape (and size!) of the cluster? I do not expect this to be without effect. I know that the answer to this question is almost impossible, but at least the problem needs to be mentioned and addressed in a convincing way.”

Response: We totally agree with the referee that the geometry (both shape and size) of deposited metals is critical to the electron transfer and the consequent catalytic reactions. Here we show the influence of M₁₃ isomers on physicochemical properties, taking the Pt₁₃/TiO₂ composite as an example, and the size effect will be discussed later in the response to the comment 5. As shown in Figure R1, the Pt₁₃ initial structure (image I) was obtained by depositing an icosahedral Pt₁₃ (the most stable structure in the gas-phase) onto the TiO₂(101) surface, which gradually evolves into a low-symmetry two-Pt-layer architecture (images II to VI). We monitored the system thermostability and tunneling barrier height (Φ_{TB} ; via hybrid HSE06 functional) during long-time AIMD simulations (> 20 ps). One can see from the upper panel that, the system energy (red dots) decreases while Φ_{TB} (blue squares) increases sharply within the first ~ 3 ps period of AIMD simulations, and the variation tendency slows down in the 3-8 ps interval and roughly reaches convergence after ~ 8 ps. In addition, we observed significant difference of ~ 3 eV in thermostability and ~ 1.7 eV in Φ_{TB} between the initial and equilibrated structures. The results explicitly signify the importance of determining stable metal/oxide interfaces on reliable calculation results.

Figure R1 Calculated system energies (red dots) and tunneling barrier heights (via HSE06 functional; blue squares) of several optimized snapshots in the AIMD trajectory of the Pt₁₃/TiO₂ composite. The geometries of six representative samples are presented in chronological order (from I to VI), showing the structural evolution of the Pt₁₃/TiO₂ composite.

Figure R2 AIMD simulation trajectories as well as the obtained optimal structures for (a) Pt₈/TiO₂(101), (b) Pt₁₃/TiO₂(101) and (c) Pt₁₉/TiO₂(101) composites. The structure equilibration generally occurs within ~8 ps in AIMD simulation as indicated by black arrows. Both the side and top views of the optimal Pt/TiO₂ structures are illustrated. Results are adopted from our previous work Figure S2 in ref. 11.

Note that all the metal/TiO₂ composites in this work were constructed based on the optimal Pt₈/TiO₂, Pt₁₃/TiO₂, Pt₁₉/TiO₂ composites determined via long-time AIMD simulations, where the time span is far longer than that required for reaching system equilibrium (20 vs. ~8 ps; Figure R2). Besides, we have also verified the good stability of the proposed Pt/Cu/TiO₂ composite again by performing AIMD simulations (>20 ps; Figure S5 in the original manuscript). All these efforts allow us to present reliable results on both the structures and energetics of metal/TiO₂ composites. We have incorporated Figures R1 and R2 as well as relevant discussions in the revised manuscript (page 5 line 2-3 and line 16-19, and page 9 line 18-20 in the text; Supplementary Fig. 4 and Supplementary Fig. 6).

“4. When H is adsorbed on the supported cluster, again there is a huge number of possible isomers. How is this specific isomer selected? To what extent the results are depending on the arbitrary choice of using these specific sites?”

Response: For HER on metals, the ΔG_H is generally regarded as the activity descriptor, and catalytic sites with too strong or endothermic ΔG_H results in low activity.^{19,20} We agree with the referee that, for metal/TiO₂ composites, differing from the extended surface, there are many different H adsorption sites on supported clusters. We have previously calculated ΔG_H results on 8~10 adsorption sites of Pt₈/TiO₂, Pt₁₃/TiO₂, Pt₁₉/TiO₂ (Figures S10-S12 and Table S4 in ref. 11) as well as the proposed bimetallic Pt₇Rh₆/TiO₂, Pt₇Cu₆/TiO₂ and Pt₆Cu₁₃/TiO₂ composites (Table S2 in the original manuscript). It was found that ΔG_H varies considerably from site to site and shows a wide energy range from 0.2 ~ -0.5 eV. Hence, under a reasonable assumption that poor surface sites basically provide minor contributions to the overall activity and also considering the volcano-shaped ΔG_H ~activity relationship,^{19,20} we uniformly selected two of the most active sites with ΔG_H approaches zero for computing the H-H coupling barrier and determining the activity. This approach allows to make fair comparisons on the surface catalytic activity of supported metal particles exposing diverse adsorption sites. Nevertheless, changes are made in the revised manuscript to make this issue clear (page 6 line 8-12 in the text).

“5. The correlation shown in Figure 4 is very interesting, but the data are vague. I count about 20 points in the plot. Are these the structures reported in Table S1? If yes, it is clear that the tunneling barrier height is strongly dependent on the cluster size: in Pt₅/TiO₂ is -2.18 eV, in Pt₁₃/TiO₂ it becomes +0.05 eV. Same metal, completely different tunnel barrier height. Not surprising, as we are in the non-scalable regime of cluster properties. Again, to what extent the results presented depend on the particular choice of the cluster size? AND CLUSTER SHAPE?”

Response: It is true that the data in Figure 4 was summarized in Table S1, and we have redesigned the plot (Figure R4b) to make it easier to understand. Regarding the significant impact of Pt particle size on the tunneling barrier height Φ_{TB} , it agrees well with our previous results that the electron transfer efficiency follows the order of $Pt_5/TiO_2 > Pt_8/TiO_2 > Pt_{13}/TiO_2 \approx Pt_{19}/TiO_2$ (Table 2 in ref. 11). These trends can be rationalized by the Bader charge results (Table S3 in ref. 11) that each Pt atom in the Pt_5 cluster is positively charged by $+0.13 |e|$ on average, showing the weakest metallic characters, while those in the Pt_{19} carry the smallest positive charge of $+0.05 |e|$ per atom. It is generally conceived that a relatively lower electron density (i.e. more positive charges) is beneficial for accepting additional electrons, which also corresponds to less intensive electrostatic potential and thus smaller Φ_{TB} . Moreover, based on the physical origins, one may expect that the size-dependent behaviours of supported metals (on TiO_2) should be a general trend not only limited to Pt. Accordingly, we here examined the size-dependent trends of supported Rh and Cu via conducting extensive AIMD simulations and IET energy calculations, as shown in Figure R3.

Figure R3 AIMD simulation trajectories (> 20 ps) as well as the side and top views of obtained optimal structures for (a) $Rh_{19}/TiO_2(101)$ and (c) $Cu_{19}/TiO_2(101)$ composites. Spin density plots (at the iso-value of $0.005 |e|/Bohr^3$) evidencing the electron transfer from TiO_2 bulk (sites I and II) to metals for (b) $Rh_{19}/TiO_2(101)$ and (d) $Cu_{19}/TiO_2(101)$ composites are also presented.

In Figure R4a we show that for all considered metals of Pt, Rh, and Cu, the electron transfer efficiency always follows the order of $M_5/TiO_2 > M_8/TiO_2 > M_{13}/TiO_2 \approx M_{19}/TiO_2$, revealing the distinct features of the same supported metal in different particle sizes. However, despite the significant impact of particle size on IET energies, we found a linear relationship between the IET energy and the tunneling barrier height (Figure R4b), irrelevant to either the metal type (indicated by different colors) or particle size (different shapes). We notice that the number of sampling points might still

be limited, but the good correlation coefficient in a wide sampling range encourages to regard such correlation as a general trend (at least for supported metals on TiO₂) that beyond the considered metal/TiO₂ models in this work. As Φ_{TB} serves as a promising electron transfer descriptor for photocatalytic reactions on metal/oxide catalysts, this finding offers possibilities to the rational design of photocatalysts via coordinating conventional catalytic descriptors (e.g., ΔG_H for HER) and the electron transfer properties Φ_{TB} . In addition, regarding the cluster shape effect, please refer to the response to the comment 3.

Figure R4 (a) Diverse IET energies of different metal particles supported on TiO₂, as well as (b) the linear correlation between IET energies and Φ_{TB} on a wide range of metal/TiO₂ composites. While Pt, Rh, and Cu are highlighted in blue, olive, and orange color, the M₅/TiO₂, M₈/TiO₂, M₁₃/TiO₂, and M₁₉/TiO₂ are illustrated by solid square, dot, triangle, and pentagon in figure (b), respectively. Symbols filled with mixed colours represent alloyed metal particles, where the stacking pattern from surface to the interface is consistent with the top-down colouring sequence.

We have incorporated Figures R3 and R4b as well as relevant discussions in the revised manuscript (page 4 line 8-10, page 7 line 1-4 from the bottom, and page 8 line 1-4 in the text; Fig. 4b and Supplementary Fig. 3).

“6. About the strong statements:

“we pioneeringly identified the tunneling barrier height as an effective electron transfer descriptor for photocatalytic reactions..”. In general should not be the authors that define their work pioneering, and furthermore there is no photocatalytic reaction studied in this paper. The sentence needs to be reformulated.”

Response: The point has been taken, and accordingly we have reformatted the sentence as ‘Moreover, we tentatively identified the tunneling barrier height as an effective descriptor for the important electron transfer process in photocatalysis on metal/oxide catalysts.’ in the revised manuscript (page 1 line 4-6 from the bottom in the text).

“7. The claim of different HER performance is based on the calculation of barriers for the H-H coupling that differ by 0.02 eV, see p. 5 (0.69 eV versus 0.71 eV). This is well below the intrinsic limitations of the methods used. Here the impression is that the authors are making too strong claims. To some extent this is true also for the effect of E_f which goes from 0.45 eV below the CB in Pt₁₃/TiO₂ to 0.65 eV for the mixed metal clusters. It is in the right direction, but not a huge change.”

Response: There seems to be some misunderstandings on the purpose of the H-H coupling results in Figure 3. Because multilayer Pt particles are thermodynamically more stable on TiO₂ surface (see ‘ $\Delta E/Pt$ ’ results in Table 1 in ref. 11), and they are good at catalyzing the H-H coupling reaction but lack of efficient electron transfer abilities, here we aimed to improve their electron transfer efficiency while maintain the robust HER activity via a function-oriented catalyst design approach. In the previous part of the manuscript, we have validated the Pt/Rh/TiO₂ and Pt/Cu/TiO₂ composites as good electron transfer materials. Subsequently, it is necessary to verify whether the catalytically efficient HER continues to proceed on the exterior Pt layer of the bimetallic composites. As pointed out by the referee, the small barrier difference between Pt/Rh/TiO₂ and Pt/Cu/TiO₂ (or relative to Pt₁₃/TiO₂) exactly demonstrates the good HER catalytic activity of both Pt/Rh/TiO₂ and Pt/Cu/TiO₂. The reason why we particularly underlined the striking results of Pt/Cu/TiO₂ is mainly because Cu has much cheaper cost than Pt and Rh, rather than drawn from the negligible barrier difference. In addition, regarding the E_f shift in alloyed metal particles, we have tuned down its contribution to the electron transfer trends in the revised manuscript (page 5 line 12-13 in the text).

“8. On the technical side, the spin densities in Figure S3 are either blue or yellow. If this indicates up and down spins, it also means that the solutions are highly spin contaminated. Some comment is in order.

In summary, a paper based on a nice idea which is not sufficiently supported and validated by the data presented.”

Response: The spin charge density plots were calculated by subtracting the charge density of spin-up states from that of the spin-down, and accordingly the yellow or blue color indicates excessive spin-polarized electrons in the system. One can see from Figure R5 that for some either pure or mixed metal particles on TiO₂, they by nature have unequal numbers of spin-up and spin-down electrons even in the charge-neutral state, consistent with literature reports.²¹⁻²³ In addition, as mentioned in the response to the comment 2, our research focused on the post-photoexcitation electron migration process, which can be approximately regarded as a ground-state event, and thus the subject does not involve the situation of spin-forbidden transition in excited-state calculations. All the electron transfer results were obtained based on the total energy change along the electron migration pathway (after photoexcitation), optimized via the DFT+U method to determine the stable electronic structure. None of any constraints on the distribution/population of spin-polarized charges were imposed during the optimization.

Figure R5 Spin density plots (at the iso-value of 0.005 |e|/Bohr³) of the charge-neutral (a) Rh₁₉/TiO₂(101), (b) Rh₁₃/TiO₂(101), (c) Pt₇Rh₆/TiO₂(101), (d) Pt₇Ir₆/TiO₂(101) and (e) Pt₇Pd₆/TiO₂(101) composites.

We have added a note in the revised manuscript to explain this issue (page 9 line 7-10 from the bottom in the text). Finally, we thank the referee again for all the positive comments and valuable suggestions.

References

- (1) Jin, H.; Li, J.; Wan, L.; Dai, Y.; Wei, Y.; Guo, H. *2D Mater.* **2017**, *4*, 025116.
- (2) Shen, T.; Ren, J. C.; Liu, X.; Li, S.; Liu, W. *J. Am. Chem. Soc.* **2019**, *141*, 3110.
- (3) Deskins, N. A.; Dupuis, M. *Phys. Rev. B* **2007**, *75*, 195212.
- (4) George, S. M. *Chem. Rev.* **2010**, *110*, 111.
- (5) Johnson, R. W.; Hultqvist, A.; Bent, S. F. *Mater. Today* **2014**, *17*, 236.
- (6) Yu, Y.; Hu, Y.; Liu, X.; Deng, W.; Wang, X. *Electrochim Acta* **2009**, *54*, 3092.
- (7) Asano, M.; Kawamura, R.; Sasakawa, R.; Todoroki, N.; Wadayama, T. *ACS Catal.* **2016**, 5285.
- (8) Liu, Y.; Gokcen, D.; Bertocci, U.; Moffat, T. P. *Science* **2012**, *338*, 1327.
- (9) Chen, R.; Bent, S. F. *Adv. Mater.* **2006**, *18*, 1086.
- (10) Wang, D.; Liu, Z. P.; Yang, W. M. *ACS Catal.* **2017**, *7*, 2744.
- (11) Wang, D.; Liu, Z. P.; Yang, W. M. *ACS Catal.* **2018**, *8*, 7270.
- (12) Di Valentin, C.; Selloni, A. *J. Phys. Chem. Lett.* **2011**, *2*, 2223.
- (13) Ma, X.; Dai, Y.; Guo, M.; Huang, B. *Langmuir* **2013**, *29*, 13647.
- (14) Zhang, Z.; Yates, J. T. *J. Phys. Chem. C* **2010**, *114*, 3098.
- (15) Thompson, T. L.; Yates, J. T. *J. Phys. Chem. B* **2005**, *109*, 18230.
- (16) Tamaki, Y.; Furube, A.; Murai, M.; Hara, K.; Katoh, R.; Tachiya, M. *Phys. Chem. Chem. Phys.* **2007**, *9*, 1453.
- (17) Tan, S.; Feng, H.; Ji, Y.; Wang, Y.; Zhao, J.; Zhao, A.; Wang, B.; Luo, Y.; Yang, J.; Hou, J. G. *J. Am. Chem. Soc.* **2012**, *134*, 9978.
- (18) Yan, L. K.; Chen, H. N. *J. Chem. Theory Comput.* **2014**, *10*, 4995.
- (19) Greeley, J.; Jaramillo, T. F.; Bonde, J.; Chorkendorff, I. B.; Norskov, J. K. *Nat. Mater.* **2006**,

5, 909.

(20) Nørskov, J. K.; Bligaard, T.; Logadottir, A.; Kitchin, J. R.; Chen, J. G.; Pandelov, S.; Stimming, U. *J. Electrochem. Soc.* **2005**, *152*, J23.

(21) Chen, H.; Li, P.; Umezawa, N.; Abe, H.; Ye, J.; Shiraishi, K.; Ohta, A.; Miyazaki, S. *J. Phys. Chem. C* **2016**, *120*, 5549.

(22) Ammal, S. C.; Heyden, A. *J. Chem. Phys.* **2010**, *133*, 164703.

(23) Zhang, S.-T.; Li, C.-M.; Yan, H.; Wei, M.; Evans, D. G.; Duan, X. *J. Phys. Chem. C* **2014**, *118*, 3514.

REVIEWERS' COMMENTS

Reviewer #1 (Remarks to the Author):

The authors have carefully revised the manuscript and properly answered all my concerns, I thus would be happy to recommend the acceptance of the paper.

Reviewer #2 (Remarks to the Author):

I read with interest the reply of the authors and I feel that the most important issues and doubts raised in the original report have been properly addressed in the revised manuscript. The paper can be recommended for publication in Nature Communications.

Reviewer #1 (Remarks to the Author):

“The authors have carefully revised the manuscript and properly answered all my concerns, I thus would be happy to recommend the acceptance of the paper.”

Response: We thank the referee for the positive comment and recommendation of manuscript publication.

Reviewer #2 (Remarks to the Author):

“I read with interest the reply of the authors and I feel that the most important issues and doubts raised in the original report have been properly addressed in the revised manuscript. The paper can be recommended for publication in Nature Communications.”

Response: We thank the referee again for the positive comments and valuable suggestions, which result in a stronger version of the manuscript.